# Protection of Fatty Liver by the Intake of Fermented Soybean Paste, *Miso*, and Its Pre-Fermented Mixture

**DOI:** 10.3390/foods10020291

**Published:** 2021-02-01

**Authors:** Ryoko Kanno, Tetsuo Koshizuka, Nozomu Miyazaki, Takahiro Kobayashi, Ken Ishioka, Chiaki Ozaki, Hideki Chiba, Tatsuo Suzutani

**Affiliations:** 1Department of Microbiology, Fukushima Medical University School of Medicine, Fukushima 960-1295, Japan; ryoko-k@fmu.ac.jp (R.K.); koshizuka-te@gifu-pu.ac.jp (T.K.); m-nozomu@fmu.ac.jp (N.M.); kobayashi4@gyosei.ac.jp (T.K.); ishiken@fmu.ac.jp (K.I.); ozaki221@fmu.ac.jp (C.O.); 2Department of Microbiology and Immunology, Gifu Pharmaceutical University, Gifu 501-1196, Japan; 3Department of Basic Pathology, Fukushima Medical University School of Medicine, Fukushima 960-1295, Japan; hidchiba@fmu.ac.jp

**Keywords:** *miso*, soy-derived foodstuffs, nonalcoholic fatty liver disease, isoflavone, leptin, resistin

## Abstract

Soybeans and fermented soy-derived foodstuffs contain many functional components and demonstrate various beneficial effects. In this report, we demonstrate the anti-fatty liver effect of *miso,* a traditional fermented product made from soybeans and rice molded in *Aspergillus oryzae* and forming a common part of the Japanese diet. After acclimation for 2 weeks, male and female C57BL/6J mice were fed with a normal diet (ND), a high-fat diet (HFD), a HFD containing 5% *miso* (HFD+M), or a HFD containing 5% pre-fermented *miso* (HFD+PFM) for 20 weeks. Although mice in the HFD group developed typical fatty liver, the consumption of *miso* or PFM significantly ameliorated the progression of fatty liver in female mice. The liver weight and the average nonalcoholic fatty liver disease activity score (NAS) were significantly reduced in the HFD+M and HFD+PFM groups. In addition, leptin and resistin levels in the serum were decreased in the HFD+M and HFD+PFM groups. The progression of fatty liver was also prevented by the consumption of *miso* or PFM in male mice, although there were no decreases in NAS. Therefore, *miso* appears to be a potential food to prevent lifestyle-related diseases such as metabolic syndrome.

## 1. Introduction

Soybean and its fermented products contain various functional materials, such as soy proteins and isoflavones, expected to have beneficial effects on our health [1,2]. Genistein and daidzein are the major isoflavones contained in soy. These isoflavones have a similar structure to that of estrogen and show weak estrogenic and anti-estrogenic activities [3]. These soy-derived isoflavones act as antioxidants and tyrosine kinase inhibitors, and prevent cardiovascular diseases, post-menopausal problems such as osteoporosis, and cancer [4,5,6]. Genistein has been shown to demonstrate anti-obesity activities both in vitro and in vivo [7]. Adipogenesis is inhibited by the addition of genistein in 3T3-L1 cells [8,9], and genistein also reduces the development of steatohepatitis in ApoE-/- mice [10]. In addition, fermentation was found to increase the serum concentration of isoflavones including genistein and daidzein [11].

Obesity is one of the most important life-style related diseases in developed countries and leads to nonalcoholic fatty liver disease (NAFLD). NAFLD has emerged as the most common chronic liver disease in developed countries due to the ongoing obesity epidemic [12,13]. Although the global prevalence of NAFLD is estimated to be 24% [14], its prevalence is growing even in the developing world [15]. NAFLD includes various liver pathologies ranging from nonalcoholic fatty liver to nonalcoholic steatohepatitis (NASH). In addition, NAFLD is a possible risk factor for hepatic failure or hepatocellular carcinoma [16,17]. Up to 25% of NAFLD patients develop NASH in their clinical course, and this can subsequently lead to cirrhosis and hepatocellular cancer [12]. Therefore, the prevention of obesity and fatty liver is a major concern for public health.

Although, several approaches are available for the management or prevention of obesity, it is most important to consume an appropriate diet, not only in terms of quantity but also content. Soybeans and soy-derived products contain many anti-obesity components including soy peptides and isoflavones; therefore, they would be effective as anti-obesity foods [18]. In this study, we evaluated *miso*, which is a Japanese traditional fermented food made from soybean paste, rice, and/or barley with salt and *A. oryzae* mold [19], as an anti-obesity food by using a mice model fed a high-fat diet (HFD). Although the consumption of HFD alone induced obesity and fatty liver in the mice, supplementation with *miso* or pre-fermented *miso* (PFM) prevented the development of fatty liver.

## 2. Materials and Methods

### 2.1. Animals and Diets

The experimental protocol of this study was approved by the Fukushima Medical University Institutional Animal Care and Use Committee. Male and female C57BL/6J mice at 3 weeks of age were purchased from Charles River Laboratories (Yokohama, Japan). After an acclimation period of 2 weeks, mice were randomly divided into 4 groups and each group was fed one of 4 diets for 20 weeks: a normal diet (ND, *n* = 6), a high-fat diet (HFD; *n* = 6), a HFD containing 5% *miso* (HFD+M, *n* = 6), or a HFD containing 5% pre-fermented *miso* (HFD+PFM, *n* = 6). ND (CE-2) and HFD (HFD-32) were purchased from CLEA Japan (Tokyo, Japan). ND contained moisture (8.84%), protein (24.85%), fat (4.06%), fiber (4.73%), ash (7.38%), and nitrogen-free extracts (49.48%). HFD contained casein (24.5%), egg white powder (5.0%), L-Cysteine (0.4%), spray-dried beef tallow (15.9%), safflower oil (20.0%), maltodextrin (8.3%), lactose (6.9%), sucrose (6.75%), vitamins (1.4%), and minerals (5.0%). The calorific contents of the diets used in this study are summarized in Table 1.

Animals were maintained in a temperature- (21 ± 2 °C) and humidity-(50 ± 20%) controlled environment with a 12 h dark–light cycle and had ad libitum access to their respective diet and water throughout the study. The body weight of the mice was measured every other week. Food intake per gauge was measured at 15, 18 and 19 weeks.

*Miso* and PFM were provided by the Central *Miso* Research Institute (Tokyo, Japan). In brief, soy paste and *A. oryzae* mold were mixed and fermented to make *miso*. PFM was simply mixed with soy paste and *A. oryzae* mold without fermentation. All animal feeds were used after γ-ray irradiation.

### 2.2. Computed Tomographic Analysis

Fat mass was evaluated at the end of the study using computed tomography (CT). The abdominal area of the mice was analyzed with an experimental animal CT system (Latheta LCT-200; Hitachi-Aloka Medical, Tokyo, Japan) under anesthesia with isoflurane (Wako Pure Chemical, Osaka, Japan). CT images were obtained from the lower end of the sternum to the upper end of the coxal bone in 96-μm slices. Quantitative assessment of the subcutaneous and visceral fat was performed using Latheta software. Subcutaneous and visceral fat tissues were distinguished by manual tracing of the abdominal wall in each of the sections. The data from 10 of 50 slices from the region of the fourth lumbar vertebra (at 5-slice intervals) were analyzed.

### 2.3. Collection of Blood and Tissue

After overnight fasting, mice were euthanized for whole blood collection by cardiac puncture under anesthesia with a cocktail of medetomidine, midazolam, and butorphanol. After measurement of its weight, the liver from each animal was fixed with 10% formalin and pathological analysis was performed on paraffin-tissue sections stained with hematoxylin–eosin and Elastica-Masson.

### 2.4. Analysis of Serum Markers

Serum glucose (GLU), triglyceride (TG), glutamic oxaloacetic transaminase (GOT), glutamic pyruvic transaminase (GPT), alkaline phosphatase (ALP), high-density lipoprotein (HDL) and total cholesterol (TCHO) were measured using a serum chemistry analyzer (Fuji Dri-chem 7000V, FujiFilm corporation, Tokyo, Japan). Serum leptin and resistin were analyzed with a Milliplex MAP kit (Merck Millipore, Burlington, MA, USA) using a Luminex 200 analyzer (Luminex, Austin, TX, USA).

### 2.5. Liver Histology

The pathological criteria for the diagnosis of nonalcoholic steatohepatitis (NASH) were evaluated by NAFLD activity score (NAS). The NAS represents the sum of steatosis, lobular inflammation, and hepatocyte ballooning scores obtained by historical analysis (range, 0–8 points, with a score of 5–8 points defined as NASH). Fibrotic stage was scored on a scale of 0–4 points as described previously [20,21].

### 2.6. UPLC Sample Preparation and Analysis

The preparation of miso and PFM extracts and UPLC analysis were performed essentially as described elsewhere [22,23]. One gram of miso or pre-fermented miso was extracted with 80% methanol at 37 °C for 2 h and then centrifuged at 1000× g for 5 min to collect 1 mL of supernatant samples. After drying in an evaporator, these supernatant samples were extracted with ethyl acetate to collect the extraction phase. The extraction phase samples were dried with nitrogen gas and dissolved in 200 µl of mobile phase consisting of 16 mM acetic acid (pH 4.6) and acetonitrile (70:30 v/v). After filtration using 0.22 µm PTFE membrane filters (Merck Millipore), samples were analyzed by ultra-performance liquid chromatography (UPLC) as described below.

The UPLC analysis was performed using an ACQUITY UPLC H-class system (Waters, Milford, MA, USA) with a turnable UV (TUV) detector. An ACQUITY UPLC^®®^ BEH C18 column (Waters, 2.1 × 100 mm, 1.7 µm) was used to analyze samples. The samples were separated using an isocratic mobile phase at a flow rate of 0.45 mL/min. Purified daidzein (Wako Pure Chemical) and genistein (Tokyo Chemical Industry, Tokyo, Japan) were used as the standards. Daidzein and genistein were monitored by UV absorption at 260 nm.

### 2.7. Statistical Analysis

Statistical differences between means were assumed to be significant at *p* values < 0.05 by one-way ANOVA followed by Tukey’s multiple comparison test using GraphPad PRISM software (GraphPad Software Inc., San Diego, CA, USA).

## 3. Results

### 3.1. Effect of Miso on Body and Adipose Tissue Weight in Mice Fed a High-Fat Diet

To evaluate the effects of *miso* in preventing obesity, male and female mice were separated into four groups: ND, HFD, HFD+*Miso* (HFD+M) and HDF+PFM. The course of body weight change in each group is presented in Figure 1. There were no significant differences in body weight among the male mice in the HFD, HFD+M and HFD+PFM groups. Although it was not significant, female mice in the HFD+M and HFD+PFM groups showed a slightly reduced body weight in comparison to those in the HFD group at the end of the experiment (Figure 1A and Table 2).

Based on the CT images, the weight of the visceral adipose tissue in the female mice was slightly reduced, although not significantly, in the HFD+M and HFD+PFM groups in comparison with those in the HFD group (Figure 2B). On the other hand, the liver weight was significantly decreased in the female HFD+M and HFD+PFM group mice but not in the male mice (Figure 2C,D and Table 3).

### 3.2. Suppression of Fatty Liver by Miso or PFM Intake

Although the HFD group developed typical fatty liver and presented a high NAS, the consumption of *miso* or PFM reduced the NAS (Table 3, Figure 3). As both the body and liver weight were increased by the consumption of HFD in female mice, liver weight per body weight was not significantly increased but the histological score was increased (Table 3). In female mice, the consumption of *miso* or PFM slightly reduced hepatic steatosis, lobular inflammation and ballooning (Figure 3A). The NAS and steatosis score were slightly decreased in the HFD+M in comparison to the HFD+PFM group in female mice, but the difference was not significant. Hepatic steatosis was increased in male mice in comparison with female mice (Figure 3B). Although the reduction in NAS in the male mice was not significant, as shown in Table 3, the pathological abnormalities in the HFD+M and HFD+PFM groups were reduced in comparison with those in the HFD group according to our pathological analysis (Figure 3B). The number of large droplets of macrovesicular fat in male mice was decreased in the HFD+M and HFD+PFM groups in comparison with that in the HFD group (Figure 3B).

### 3.3. Serum Biochemical Markers of Fatty Liver and Adipokines

To evaluate the development of fatty liver, serum biochemical markers were analyzed as shown in Table 4. The increase in biochemical markers in the HFD groups, especially GOT, GPT, and ALP, indicate liver damage and provide supporting information on the development of NAFLD. Among the biochemical markers, total cholesterol (TCHO) and GPT were significantly reduced in female mice in the HFD+M and HFD+PFM groups, respectively. The ALP score for female ND mice was slightly higher than that for HFD mice, probably due to hemolysis or individual variations. Although there were no significant differences, GLU, ALP, TG and HDL all showed a tendency to be lower in female mice with the consumption of *miso* or PFM. These results indicate the prevention of liver function deterioration by *miso* or PFM. In male mice, GPT showed a tendency to decrease with the consumption of *miso* or PFM, but again this decrease was not significant. These results suggested that the consumption of *miso* or PFM prevented liver cell damage.

The serum levels of adipokines, leptin and resistin, were increased by the consumption of HFD as the mice developed obesity (Figure 4). In female mice, both leptin and resistin levels were decreased by the consumption of *miso*. Although the serum resistin level was clearly decreased, the serum leptin level was lowered, but not significantly so, by the consumption of HFD+PFM. On the other hand, no significant changes in serum leptin or resistin levels were detected in the male mice.

### 3.4. The Daidzein and Genistein Content in Miso or Pre-fermented Miso

In order to evaluate the amount of soy-derived isoflavones, daidzein and genistein, in *miso* or pre-fermented *miso*, samples were analyzed by UPLC as described in Materials and Methods. As shown in Table 5, both daidzein and genistein were detected in the *miso* and PFM, and the content was increased by fermentation. Figure 5 shows the chromatogram of standards (A), the extracts of *miso* (B), and the extracts of PFM (C).

## 4. Discussion

*Miso* is a traditional foodstuff in Japan and has many biological benefits [19]. In this report, we showed that *miso* and PFM have some beneficial effects in preventing the development of fatty liver disease, which is induced by the consumption of high-fat diets in mice. In female mice, in particular, the consumption of *miso* and PFM clearly reduced liver weight and NAS (Figure 2C and Figure 3). In addition, the weight of visceral fatty tissue was reduced by the intake of *miso* and PFM (Figure 2B). Both the intake of fermented and pre-fermented *miso* prevented the progression of fatty liver, indicating that the mold of soy-paste and *A. oryzae* contained beneficial components. *Miso* is a typical traditional Japanese food and about 70% Japanese people take *miso* soup every day [24]. Okada et al. also presented the data that Japanese food intakes reduce all-cause cardio vesicular disease, and cancer mortality among the Japanese population [24]. However, the prevalence of NAFLD is becoming more common in Japanese population as in other developed countries [25,26], further examination is necessary to clarify the effect of *miso* against NAFLD.

The progression of fatty liver in female mice was clearly prevented by the intake of *miso* or PFM (Figure 3). Our results indicate that soy-derived isoflavones, which are contained in *miso* and PFM as shown in Table 5, could be potent components in the ameriolation of fatty liver in our model. Isoflavones and proteins from soy demonstrate many beneficial effects against cancer, menopause, osteoporosis, cardiovascular and metabolic problems [27]. Among the isoflavones present in soybeans, genistein has shown therapeutic potential against NAFLD in both in vitro and in vivo models [28]. The subcutaneous injection of genistein in rats fed HFD reduces liver steatosis and inflammation [29], and dietary supplementation with genistein prevents liver steatosis and adipose tissue dysfunction in mice [30]. Genistein is a candidate for the functional component preventing liver steatosis as it is present in *miso* even after fermentation [31]. The derivative form of genistein has anti-inflammatory activities and downregulates iNOS and COX-2 expression in RAW264.7 cells [32].

The effects of *miso* and PFM were more significant in female mice (Figure 3). Sex-dependent differences have also been reported in the beneficial effects derived from isoflavones [33]. The elimination rate of genistein is faster in males than females [34]. In addition, female hormone signaling might be involved in the metabolism of isoflavones as the bioavailability of isoflavone is reduced in ovariectomized female mice [35]. Although, the relationship with the consumption of *miso* is unclear, the prevalence of NAFLD in women is lower than men in Japan [36]. Further investigation is required to elucidate the sex-dependent effects of genistein.

The results presented herein indicate a reduction in serum leptin and resistin levels in female mice by the consumption of *miso* or PFM (Figure 4A,B). Both leptin and resistin are important adipokines that regulate insulin sensitivity in rodents [37]. Leptin is an insulin-sensitizing hormone secreted from adipose tissue and regulates body weight and fat mass. The reduction in leptin levels indicates a decrease in adipose tissue and the prevention of obesity. Resistin is also secreted from adipose tissue and was shown to be involved in insulin resistance in a rodent model [38]. The reductions in leptin and resistin in our model correlate with the reduction in adipose tissues presented in the CT images (Figure 2). These results indicate that amelioration of obesity by the intake of *miso* or PFM.

NAFLD, which is caused by obesity, is the most common liver disease and is a serious health-care issue worldwide [39,40]. NAFLD includes various liver pathologies ranging from nonalcoholic fatty liver to nonalcoholic steatohepatitis (NASH). NASH leads to liver cirrhosis, hepatocellular carcinoma and liver failure without excessive alcohol intake [41]; therefore, the prevention of obesity is an important issue for public health. A number of clinical studies have shown that soy and its derived products have some beneficial effects in preventing the development of obesity [42,43,44]. Our results are basically consistent with these clinical results and indicate the usefulness of soy-derived foodstuffs.

## 5. Conclusions

Our results indicate that *miso* and its pre-fermented product are functional foodstuffs that can prevent the progression of obesity and fatty liver. In particular, the consumption of *miso* ameliorated the progression of fatty liver and reduced serum adipokines in female mice. The applicability of *miso* as a functional foodstuff is likely to be increased by the elucidation of these mechanisms and the underlying functional component.

## Figures and Tables

**Figure 1 foods-10-00291-f001:**
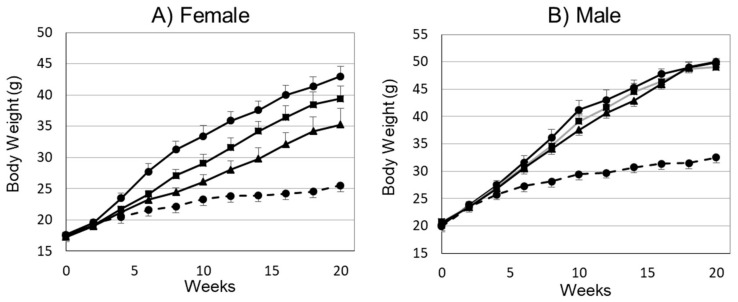
Average body weight of female (**A**) and male (**B**) mice. Normal diet (ND): dotted line; HFD: closed circles; HFD+ *miso* (M): closed squares; HFD+ pre-fermented *miso* (PFM): closed triangles. Error bars indicate the standard error of mean (SEM) (*n* = 5 or 6).

**Figure 2 foods-10-00291-f002:**
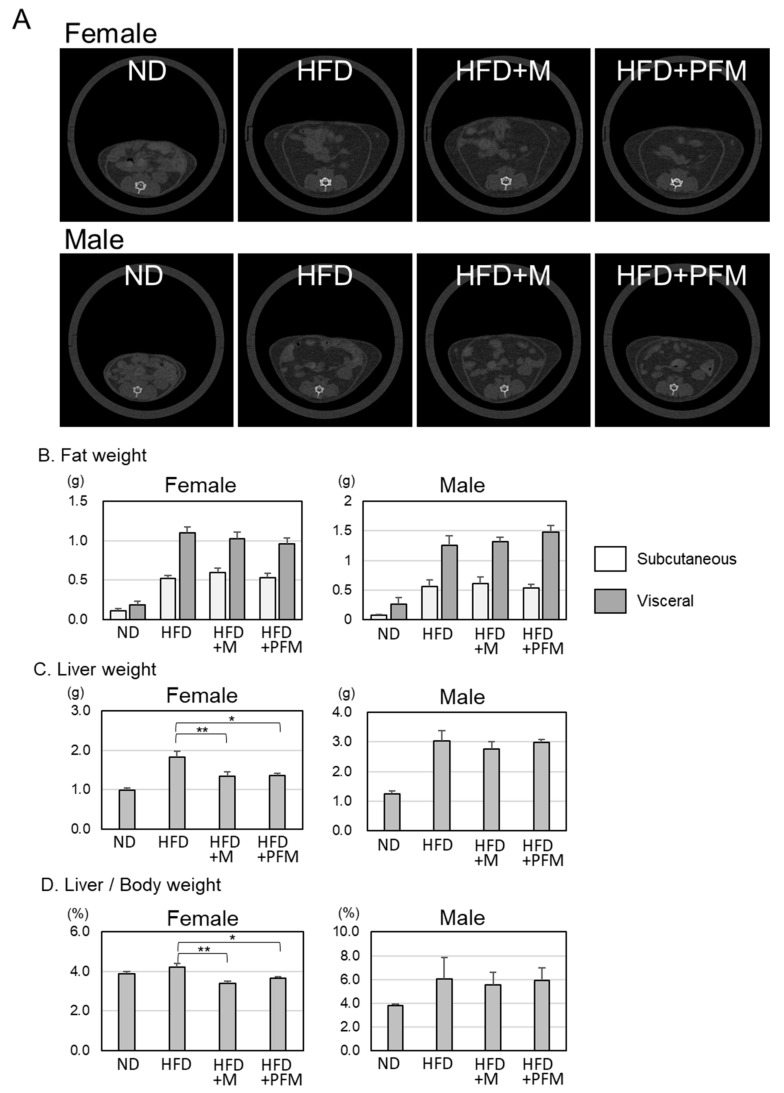
The reduction in visceral fatty tissue and liver weight in female mice by the consumption of *miso* or PFM. (**A**) Representative CT abdominal images of male and female mice used for quantification of abdominal fat deposits. (**B**) The area of adipose tissue was measured and converted to weight with Latheta software. White and gray bars indicate subcutaneous and visceral adipose tissue weight, respectively. (**C**,**D**) The liver weight (**C**) and liver/body weight (**D**) of mice. Error bars indicate SEM (*n* = 5 or 6). Values with different superscripts are significantly different among HFD groups by ANOVA with Tukey’s multiple comparison test. * *p* < 0.05; ** *p* < 0.01.

**Figure 3 foods-10-00291-f003:**
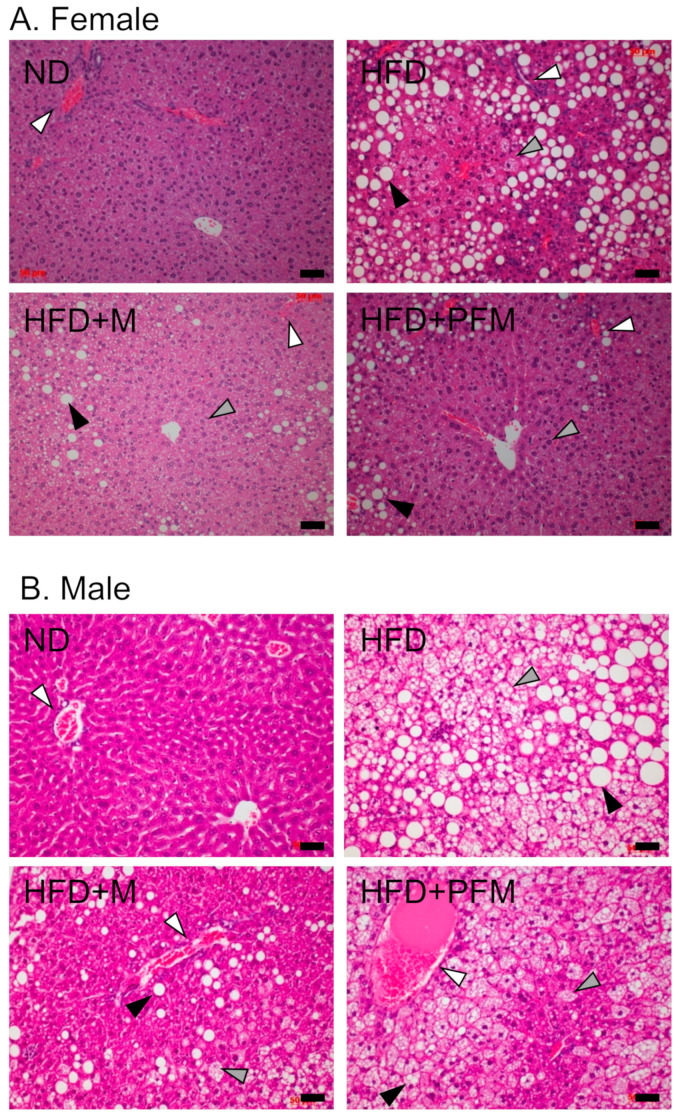
Hematoxylin and eosin (H&E) staining of liver tissue sections. Liver tissue sections from the female (**A**) and male (**B**) groups are shown. The normal diet group (ND), high-fat diet group (HFD), HFD containing *miso* group (HFD+M) and HFD containing pre-fermented *miso* group (HFD+PFM) are shown. White, gray, and black arrowheads indicate portal vein, micro-vesicles, and macro-vesicles, respectively. Bars = 50 μm.

**Figure 4 foods-10-00291-f004:**
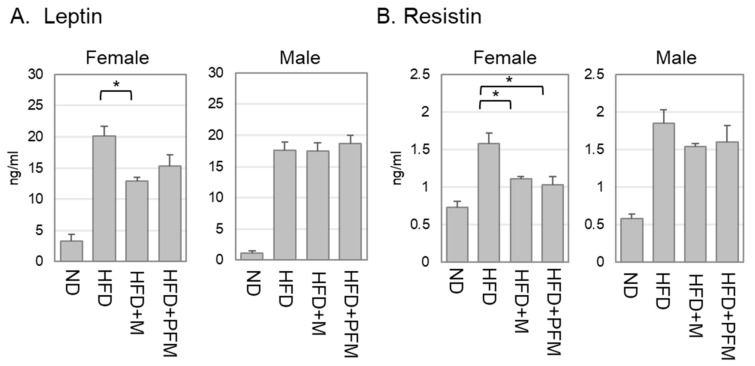
The reduction in leptin and resistin levels by the consumption of *miso* and PFM. The levels of leptin (**A**) and resistin (**B**) in the serum from mice. Error bars indicate S.E. (*n* = 5 or 6). Values with different superscripts are significantly different among HFD groups by ANOVA with Tukey’s multiple comparison test. * *p* < 0.05.

**Figure 5 foods-10-00291-f005:**
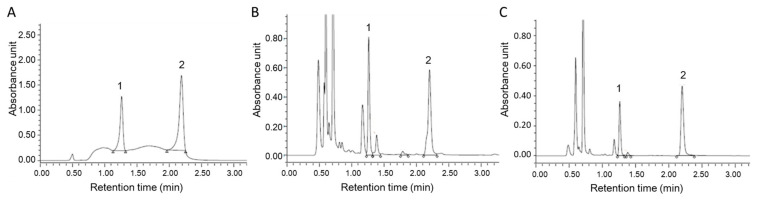
UPLC chromatograms of (**A**) standards, (**B**) extracts from *miso* and (**C**) extracts from PFM monitored by UV (260 nm). Peak 1: Daidzein; peak 2: Genistein.

**Table 1 foods-10-00291-t001:** Calorific contents of diets used in this study.

	ND ^1^	HFD ^2^	HFD+M ^3^	HFD+PFM ^4^
Protein (%)	24.85	26.45	25.98	25.64
Carbohydrate (%)	49.48	26.23	25.64	26.6
Fat (%)	4.06	33.41	32.26	32.03
Energy (kcal/100g)	333.8	511.4	496.7	497.4

^1^ CE2 diet; ^2^ a high-fat diet (HFD) 32 diet; ^3^ HFD32 containing 5% *miso*; ^4^ HFD32 containing 5% pre-fermented *miso*.

**Table 2 foods-10-00291-t002:** Body weight gain and body composition between groups.

	Female				Male			
**Groups**	**ND**	**HFD**	**HFD+M**	**HFD+PFM**	**ND**	**HFD**	**HFD+M**	**HFD+PFM**
Initial body weight (g)	17.5 ± 0.74	17.7 ± 0.31	17.6 ± 0.27	17.2 ± 0.26	19.9 ± 0.89	20.1 ± 0.51	20.7 ± 0.55	20.5 ± 0.29
Final body weight (g)	25.5 ± 0.78	43.0 ± 1.62	39.4 ± 2.07	37.5 ± 2.59	32.5 ± 0.72	49.9 ± 0.76	49.0 ± 1.41	50.0 ± 1.11
Body weight gain (g)	8.0 ± 0.78	25.4 ± 1.45	21.9 ± 2.07	20.3 ± 2.61	12.6 ± 0.76	29.8 ± 0.64	28.3 ± 0.97	29.6 ± 0.93
Feed intake (g/day) ^1^	3.3 ± 0.06	2.7 ± 0.05	2.6 ± 0.06	2.7 ± 0.07	3.4 ± 0.10	2.8 ± 0.05	2.9 ± 0.08	3.0 ± 0.06
Energy intake (kcal/day)	11.0 ± 0.21	13.8 ± 0.28	12.9 ± 0.31	13.4 ± 0.34	11.3 ± 0.33	14.3 ± 0.27	14.4 ± 0.40	14.9 ± 0.29

^1^ Average of the food intake at 15, 18 and 19 weeks. All values are mean ± SEM.

**Table 3 foods-10-00291-t003:** Liver weight and nonalcoholic fatty liver disease (NAFLD) activity score (NAS).

	Female				Male			
	**ND**	**HFD**	**HFD+M**	**HFD+PFM**	**ND**	**HFD**	**HFD+M**	**HFD+PFM**
Liver weight (g)	1.0 ± 0.05	1.8 ± 0.14	1.3 ± 0.11 ^†^	1.4 ± 0.05 *	1.23 ± 0.04	3.04 ± 0.41	2.75 ± 0.28	2.98 ± 0.28
Liver weight (%) ^1^	3.9 ± 0.11	4.2 ± 0.17	3.4 ± 0.12 ^†^	3.7 ± 0.06 *	3.8 ± 0.07	6.0 ± 0.74	5.6 ± 0.42	5.9 ± 0.44
NAS	0.83 ± 0.14	5.67 ± 0.51	3.33 ± 0.19 ^†^	4.00 ± 0.28 *	0.67 ± 0.19	6.0 ± 0.00	5.83 ± 0.15	5.5 ± 0.31
Steatosis	0.00 ± 0.00	2.16 ± 0.28	1.33 ± 0.19 *	1.80 ± 0.18	0.00 ± 0.00	3.0 ± 0.00	2.83 ± 0.15	2.83 ± 0.15
Lobular inflammation	0.83 ± 0.15	1.66 ± 0.19	1.00 ± 0.00 *	1.00 ± 0.00 *	0.67 ± 0.19	1.0 ± 0.00	1.0 ± 0.00	0.83 ± 0.15
Ballooning	0.00 ± 0.00	1.83 ± 0.15	1.00 ± 0.00 ^†^	1.20 ± 0.17 *	0.00 ± 0.00	2.0 ± 0.00	2.0 ± 0.00	1.83 ± 0.15

All values are mean ± SEM. Values with different superscripts are significantly different among HFD groups by ANOVA with Tukey’s multiple comparison test. * *p* < 0.05; ^†^
*p* < 0.01; ^1^ liver weight per body weight.

**Table 4 foods-10-00291-t004:** Effects of *miso* or PFM on serum levels of GLU, GOT, GPT, ALP, TCHO, TG and HDL.

	Female				Male			
	**ND**	**HFD**	**HFD+M**	**HFD+PFM**	**ND**	**HFD**	**HFD+M**	**HFD+PFM**
**GLU**	122.1 ± 20.8	303.8 ± 17.5	269.6 ± 33.8	248.8 ± 31.1	247.3 ± 34.4	492.0 ± 55.4	492.5 ± 42.2	602.2 ± 66.4
**GOT**	285.3 ± 112.7	370.8 ± 46.0	371.0 ± 54.5	418.5 ± 94.0	105.5 ± 15.5	105.5 ± 43.3	184.3 ± 24.0	146.7 ± 11.7
**GPT**	38.1 ± 3.1	187.0 ± 19.7	111.8 ± 25.0	93.5 ± 14.4 *	28.8 ± 3.1	156.3 ± 41.2	119.0 ± 19.9	72.5 ± 9.6
**ALP**	290.6 ± 15.5	271.1 ± 15.5	242.0 ± 6.8	204.6 ± 22.8	252.1 ± 11.0	337.1 ± 38.1	328.8 ± 20.7	284.0 ± 13.0
**TCHO**	79.8 ± 5.0	110.3 ± 4.5	83.8 ± 8.7^*^	89.6 ± 4.0	86.8 ± 3.8	187.3 ± 18.0	186.1 ± 6.9	169.7 ± 5.3
**TG**	84.5 ± 14.0	38.8 ± 5.3	36.0 ± 5.9	31.6 ± 5.0	88.0 ± 5.7	59.3 ± 2.8	46.3 ± 3.4	42.2 ± 2.7
**HDL**	70.6 ± 5.1	108.3 ± 5.3	82.3 ± 10.2	88.6 ± 3.2	76.6 ± 4.1	192.6 ± 19.4	191.3 ± 7.1	170.5 ± 7.4

GLU: glucose; GOT: glutamic oxaloacetic transaminase; GPT: glutamic pyruvic transaminase; ALP: alkaline phosphatase; TCHO: total cholesterol; TG: triglyceride: HDL: high-density lipoprotein. All values are mean ± SEM. Values with different superscripts are significantly different among HFD groups by ANOVA with Tukey’s multiple comparison test. * *p* < 0.05.

**Table 5 foods-10-00291-t005:** The daidzein and genistein content in *miso* or pre-fermented *miso.*

	*Miso*	PFM
**Daidzein (µg/g)**	262.66	91.37
**Genistein (µg/g)**	192.58	116.82

## Data Availability

The datasets presented in this study are available on request to the corresponding author.

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
