# Peer review of "Protection of Fatty Liver by the Intake of Fermented Soybean Paste, Miso, and Its Pre-Fermented Mixture"

_foods, 2021, doi:10.3390/foods10020291_

Round 1
Reviewer 1 Report
The study conducted by Ryoko Kanno et al., focuses on evaluated miso, which is a Japanese traditional fermented food as an anti-obesity food by using a mice model fed a high-fat diet (HFD). This is an important study given life-style related diseases in developed countries are increasing annually. Below are the reviewer’s critiques
Results
Table 2: Explain why authors chose to following weeks 15, 18 and 19 to determine the average of the food intake.
Figure 2: Present CT abdominal images for male mice along with female mice
Figure 3: Use arrow/start etc. to indicate histological changes (portal vein, micro- and macro-vesicles etc.) in the histopathological images.
Figure 4: Combine A&C into one figure and B&D into another figure.
Discussion
Discuss why some studies presented data that it is more effective in males [32, 33]. Include factors that might influence the outcome.
Author Response
Responses to the Reviewer’s comments
Manuscript ID: foods-1071780
We thank you for your review on our manuscript. To accommodate your comments, we modified figure 2, 3, and 4 and revised the text as described below. Line numbers indicated on the revised manuscript without tracked changes.
Reviewer 1
Point 1:
Table 2: Explain why authors chose to following weeks 15, 18 and 19 to determine the average of the food intake.
Response:
Since the body weight of mice were constantly increased, we determined the food intakes in these terms.
Point 2:
Figure 2: Present CT abdominal images for male mice along with female mice.
Response:
As requested, the CT images, fat weight, liver weight, and liver/body weight data of male mice were shown in Figure 2. Sentences were modified in lines 150-153.
Point 3:
Figure 3: Use arrow/start etc. to indicate histological changes (portal vein, micro- and macro-vesicles etc.) in the histopathological images.
Response:
As suggested, portal vein, micro-, and macro-vesicles were shown as arrowheads. The portal vein was not clear in male HFD groups. Sentence was added in line 191-192.
Point 4:
Figure 4: Combine A&C into one figure and B&D into another figure.
Response:
Thank you for your advice. As suggested, Figure 4 was revised.
Point 5:
Discussion
Discuss why some studies presented data that it is more effective in males [32, 33]. Include factors that might influence the outcome.
Response:
Since there was equivocally in the description, sentence and references were removed. However, our conclusion, "In female mice, in particular, the consumption of miso and PFM clearly reduced liver weight and NAS", was not altered.
Reviewer 2 Report
GENERAL COMMENT
The chief finding of the present study is that either miso - a traditional Japanese food derived from fermented soybeans and rice – or pre-fermented miso (PFM) significantly ameliorated the progression of fatty liver in female mice. The progression of fatty liver was also prevented by the consumption of miso or PFM in male mice, although there were no decreases in NAS. This is an important result given that NAFLD is prevalent among genral population and lacks adequate therapeutic strategies. Additionally, I became interested in noting sexual dimorphism of this effect, which further emphasizes the eminently sexually dimorhphic nature of NAFLD. This must be more extensively discussed.
SPECIFIC COMMENT
MAJOR
"NAFLD represents a wide spectrum of clinical entities from asymptomatic hepatic steatosis to more advanced liver disease with hepatic failure or hepatocellular carcinoma " --> A supporting reference must be added here. In addition, it would sem logocal to link the above statement with the following "NAFLD includes various liver pathologies ranging from non-alcoholic fatty liver to non-alcoholic steatohepatitis (NASH). " In doing so, based on th efndings of the present study, it ia slo important to highlight that sex is a major modifier of NAFLD risk of development and progression.
Throughout the manuscript reord "non-alcoholic" to "nonalcoholic such as originally proposed by Ludwig, Shaffner and Thaler who were first in coining this adjective.
"Several approaches are available for the management or prevention of obesity, including bariatric surgery, exercise, drugs and vaccines " Please, note that references [12, 16] do not adequately support this statement and must be reworked.
The main finding of the present study was that "In female mice, in particular, the consumption of miso and PFM clearly reduced liver weight and NAS". I think these Authors may be willing to further comment on this important result which is in agreement with the acknowledged sexual dimorphism of NAFLD.
These Authors may be willing to discuss whether they believe that thanks to miso and PFM traditional Japaenese diet may be a valid adjunt to NAFLD prevention: are there any epidemiological data supporting such a contention ?
MINOR
“liver steatohepatitis” please note that this is a much redundant statement given that “steatohepatitis” already means “fatty inflammation of the liver” and, therefore does not need to be further qualified as “liver”
Author Response
Responses to the Reviewer’s comments
Manuscript ID: foods-1071780
We thank you for your review on our manuscript. Line numbers indicated on the revised manuscript without tracked changes.
Reviewer 2
Point 1:
"NAFLD represents a wide spectrum of clinical entities from asymptomatic hepatic steatosis to more advanced liver disease with hepatic failure or hepatocellular carcinoma " --> A supporting reference must be added here. In addition, it would sem logocal to link the above statement with the following "NAFLD includes various liver pathologies ranging from non-alcoholic fatty liver to non-alcoholic steatohepatitis (NASH). " In doing so, based on the findings of the present study, it is also important to highlight that sex is a major modifier of NAFLD risk of development and progression.
Response:
We would like to explain the possible risk of NAFLD in this sentence. NAFLD leads to severe disease such as hepatic failure or hepatocellular carcinoma. Therefore, we would like to modify and move this sentence to lines 48-49 and added new references (Kulik et al., 2019 PMID: 30367835 as reference #16, Marengo et al., 2016 PMID: 26473416 as reference #17). In addition, please see the responses to comment 4 and 5 about sex difference and prevalence of NAFLD.
Point 2:
Throughout the manuscript reord "non-alcoholic" to "nonalcoholic such as originally proposed by Ludwig, Shaffner and Thaler who were first in coining this adjective.
Response:
As suggested, we modified this word in line 24 and thereafter.
Point 3:
"Several approaches are available for the management or prevention of obesity, including bariatric surgery, exercise, drugs and vaccines " Please, note that references [12, 16] do not adequately support this statement and must be reworked.
Response:
Since there is a lot of reports about the method to prevent obesity, such as bariatric surgery or exercise, we couldn’t narrow down the adequate reference. In this sentence, we would like to emphasis the importance of quality of nutrients to management or prevention of obesity. Therefore, the sentences were modified as lines 53-54.
Point 4:
The main finding of the present study was that "In female mice, in particular, the consumption of miso and PFM clearly reduced liver weight and NAS". I think these Authors may be willing to further comment on this important result which is in agreement with the acknowledged sexual dimorphism of NAFLD.
Point 5:
These Authors may be willing to discuss whether they believe that thanks to miso and PFM traditional Japanese diet may be a valid adjunt to NAFLD prevention: are there any epidemiological data supporting such a contention?
Response 4 and 5:
We appreciate your important comments to improve our manuscript.
In Japan, about 70% people take miso soup every day (Okada et al., 2018. Br. J. Nutri. PMID: 29923480 as reference #24). In addition, according to the data of health checkup, the prevalence of NAFLD is lower in female in Japan (Eguchi et al., 2012. J. Gastroenterol. PMID: 22328022 as reference #36). Although, the direct relationships were not clear, these reports may support our findings. However, in recently, the prevalence of NAFLD is become more common in Japanese population as other developed countries (Eguchi et al., 2020. JGH Open. PMID: 33102749 as reference #25, Estes et al., 2018. J. Hepatol. PMID: 29886156 as reference 26), further examination is necessary to clarify the effect of miso against NAFLD.
Sentences were added in lines 228-234 and 251-252.
Minor comment
Line 39. As suggested, “liver” was removed.